# Psychological Impact of Pro-Anorexia and Pro-Eating Disorder Websites on Adolescent Females: A Systematic Review

**DOI:** 10.3390/ijerph18042186

**Published:** 2021-02-23

**Authors:** Carmela Mento, Maria Catena Silvestri, Maria Rosaria Anna Muscatello, Amelia Rizzo, Laura Celebre, Martina Praticò, Rocco Antonio Zoccali, Antonio Bruno

**Affiliations:** 1Department of Biomedical, Dental Sciences and Morphofunctional Imaging, University of Messina, Psychiatric Unit Policlinico “G. Martino” Hospital, 98124 Messina, Italy; mmuscatello@unime.it (M.R.A.M.); lallacelebre@gmail.com (L.C.); martinapratico@hotmail.com (M.P.); rocco.zoccali@unime.it (R.A.Z.); antonio.bruno@unime.it (A.B.); 2Psychiatric Unit, Policlinico Hospital “G. Martino”, 98124 Messina, Italy; mariacate@libero.it (M.C.S.); amrizzo@unime.it (A.R.)

**Keywords:** eating abnormal behavior, pro-ana and pro-mia websites, female adolescents

## Abstract

(1) Background: Teenagers (in particular, females) suffering from eating disorders report being not satisfied with their physical aspect and they often perceive their body image in a wrong way; they report an excessive use of websites, defined as PRO-ANA and PRO-MIA, that promote an ideal of thinness, providing advice and suggestions about how to obtain super slim bodies. (2) Aim: The aim of this review is to explore the psychological impact of pro-ana and pro-mia websites on female teenagers. (3) Methods: We have carried out a systematic review of the literature on PubMed. The search terms that have been used are: “*Pro*” AND “*Ana*” OR “*Blogging*” AND “*Mia*”. Initially, 161 publications were identified, but in total, in compliance with inclusion and exclusion criteria, 12 studies have been analyzed. (4) Results: The recent scientific literature has identified a growing number of Pro Ana and Pro Mia blogs which play an important role in the etiology of anorexia and bulimia, above all in female teenagers. The feelings of discomfort and dissatisfaction with their physical aspect, therefore, reduce their self-esteem. (5) Conclusion: These websites encourage anorexic and bulimic behaviors, in particular in female teenagers. Attention to healthy eating guidelines and policies during adolescence, focused on correcting eating behavioral aspects, is very important to prevent severe forms of psychopathology with more vulnerability in the perception of body image, social desirability, and negative emotional feedback.

## 1. Introduction

Eating disorders are multifactorial disorders, affecting about 0.3% of female teenagers; they have a prevalence included between 1.2% and 4.2%. In particular, anorexia nervosa arises during adolescence, and it can be devastating, potentially included suicidal risk [1,2,3]. A common trait among young women with eating disorders is low self-esteem, with a tendency towards depressed moods. Many factors seem to influence the development and preservation of these disorders among female teenagers. The scientific panorama shows how teenagers with eating disorders have a distorted body image, an wrong perception of their image, and therefore, are more dissatisfied with their physical aspect, in comparison to celebrities. Body image can generate feelings of satisfaction or dissatisfaction and this can cause the individual to make radical choices when treating his/her body [4,5]. In this context, there are a variety of pro-eating disorder communities (websites), and teenagers use social media to talk about their physical aspect, their activities, and to exchange advice about weight loss; this condition supports anorexia nervosa, as the problem of weight loss becomes relevant for their lives and the solution to their health problems. In fact, according to the scientific literature, all members of these communities have reported high levels of eating disorders [6].

Pro-ana and pro-mia websites are virtual spaces, in which teenagers can exchange ideas about their body image and physical aspect. An uncontrolled use of these websites is common practice among teenagers, in particular among young women, and is a factor related to eating disorders. In this regard, some years ago, several authors started to include addiction to the Internet and eating disorders among problematic behaviors in adolescents. Frequent use of social networks and the possibility to follow celebrities (e.g., influencers, models, actors, and actresses) influence perception of the self and of one’s own physical and psychological way of being. This happens when there is a social meeting that can influence the user’s (teenager) mood. This desire to look like celebrities on social networks can promote the insurgence and/or preservation of eating disorders such as anorexia and bulimia. The influence of social media is growing among female teenagers with advice and tips to lose weight and showing images of thin bodies.

In line with different studies, looking at images of underweight celebrities is associated to an ideal body image and aspiration to lose weight and these conditions can promote eating disorders [7,8].

Pro-ana and pro-mia websites offer feedback on people’s physical aspect: teenagers receive comments on their aspect and advice on how to lose weight [9,10,11,12,13,14,15,16,17]. These forums are private and the topic being dealt with is the philosophy of absolute thinness [18,19,20]. In recent years, a study by Almenara and colleagues [21] demonstrated that looking for sensations and disinhibition online were both associated with a higher risk of exposure to ana–mia websites for teenage women and young adults. According to Borzekowski et al. (2010), most websites (58%) contain images of celebrities with ultra slim bodies and promote an anorexic lifestyle, and teenagers visiting pro-ana websites seem to have higher levels of body dissatisfaction and eating disorders [10].

Bates (2015) examined the metaphors used in a pro-ana website to talk about oneself. The author has applied the Metaphor Identification Procedure to 757 text profiles and has identified four key metaphorical constructions in self-description by pro-ana members: self as space, self as weight, and improving the self and social self. Bert and colleagues (2016) found 341 pro-ana accounts on Twitter; for each account, the authors analyzed the number of followers, users’ biographical information, and have studied the most used hashtags. These accounts were very popular, having 23.609 followers; users were mainly young women (97.9 percent) and teenagers. This study demonstrated that the most used hashtags were: “thinspiration”, “proana”, “thin15”, “’ana tips”. These accounts contain dangerous information about body image, eating habits, and physical aspect. Bragazzi et el. (2019) studied 402 websites and demonstrated that the media tend to spread images of models who are abnormally thin.

In a previous study, Çelik and colleagues (2015) found a positive correlation between problematic use of the Internet and approach to food, and the problematic use of the Internet predicted these approaches [22]. The dimension of the effect in relation to the negative impact of pro-ED websites is related to eating pathology. Several authors have counted a large number of websites promoting dysfunctional eating behaviors such as anorexia and bulimia [23,24,25,26].

In light of the collected data in the scientific literature, the aim of this report is to explore the psychological impact of pro-ana and pro-mia websites on female teenagers.

## 2. Materials and Methods

Data for this systematic review have been collected in compliance with the reporting elements used for systematic reviews and meta-analysis [27]. PRISMA consists of a checklist aimed at making preparation and reporting of review/meta-analysis studies easier, by identifying, selecting, and critically assessing analyzed research and analyzing data from the studies included in the review.

### 2.1. Criteria for Eligibility

The articles have been included in the review according to the following inclusion criteria: English language, publication in peer reviewed journals, quantitative information on language processing in movement disorders, and year of publication from 2015—2020. Articles have been excluded according to title, abstract, or complete text for the processes linked to the psychological impact of pro-ana and pro-mia websites on adolescents and to irrelevance to the topic being dealt with. Further exclusion criteria were review articles, editorial comments, and case reports/series. Moreover, it was arbitrarily decided to start our research in 2015 to provide a more recent outlook on the psychological impact of pro-ana and pro-mia websites among teenagers.

### 2.2. Research Strategy

This systematic review has been carried out according to systematic review guidelines [27]. The PubMed database has been searched from 1 January 2015 to 1 January 2020, using 4 key terms related to this topic (“Pro” AND “Ana” OR “Blogging” AND “Mia”). The electronic research strategy used for PubMed is described in Table 1. The articles have been selected according to title and abstract; the entire article has been read if the title/abstract was related to the specific issue of the psychological impact of pro-ana and pro-mia websites on female teenagers and if the article potentially met inclusion criteria. Moreover, references to the selected articles have been examined in order to identify further studies which could meet the inclusion criteria. 

### 2.3. Search Strategy and Study Selection

Overall, bibliographical research has been carried out in the PubMed database, with final research updated to January 2020. Initial research used key terms “Pro” AND “Ana” OR “Blogging” AND “Mia”. The key terms used are related to processes connected to the psychological impact of pro-ana and pro-mia websites or blogs.

Figure 1 sums up the flow chart of the articles selected for review. Research in the PubMed database provided a total of 161 quotations; no additional studies meeting the inclusion criteria were identified when checking the reference list of the selected documents.

After checking duplicate copies, 61 records have been examined. Among these, 28 studies have been excluded, based on inclusion and exclusion criteria. After the screening, a total of 12 studies, having checked the processes connected to the psychological impact of pro-ana and pro-mia websites on female teenagers, have met the inclusion criteria and have been included in the systematic review (Table 2).

The selected studies have demonstrated the relationship between pro-anorexia and pro-bulimia websites/forums and eating behaviors; a variety of studies have dealt with this topic, in particular research on the relation between pro-ana and pro-mia websites and the desire to lose weight.

This review has examined the psychological impact of pro-ana and pro-mia websites on female teenagers. Articles have been selected according to title and abstract; the entire article has been read if the title/abstract was related to the specific issue of the psychological impact of pro-ana and pro-mia websites on female teenagers in relation to eating disorders; and if the article potentially met the inclusion criteria. Details are reported in Table 1 and Table 2. Additionally, references to the selected articles have been examined in order to identify further studies which could meet inclusion criteria (Table 3). 

### 2.4. Bias Risk among Studies 

In all studies included in this review, a potential bias of the database should be considered. Only articles in English have been used, which could have compromised access to articles published in other languages.

## 3. Results

Most analyzed research has focused on the negative influence of websites in female teenagers with eating disorders such as anorexia and bulimia. These websites have seemed to offer a sense of support to teenagers vulnerable to eating disorders. These studies have explored teenagers’ exposure to these websites, personal profiles related to popular access to social network, as well as pro-ana accounts on Twitter [18,21,26,28]. Other more social aspects, linked to communication and language, have been explored in a recent study on language and information used on this website [29,30,31]. The relationship between a problematic use and abuse of the Internet and eating behaviors in adolescents has been investigated, as well as negative online support in the case of pro-anorexia websites [11,14,22]. Psychological aspects are generally explored as a potential risk of eating disorders to exacerbate symptoms of users’ websites eating disorders in a sample population [17,32]. Another study has explored the physical and mental state of people participating in pro-anorexia web communities [33]. In particular, Almenara and colleagues (2016) have demonstrated that looking for sensations and disinhibition online were both associated with a higher risk of exposure to ana–mia websites, in male and female teenagers, although some gender differences were evident. In girls, but not among boys, the older the teenager was and the higher her socioeconomic status was, the higher the chances were of being exposed to “ana-mia” websites. Bates (2015) has identified four key metaphorical constructions in self-description by pro-ana members: self as space, self as weight, and improving the self and social self. These four main metaphors represented speech strategies, both in order to create a collective pro-ana identity and to enact an individual identity as pro-ana. Bert et al. (2016) have highlighted a high number and popularity pro-anorexia groups on Twitter. These accounts contain dangerous information, especially considering the users’ young ages. The investigation by Bragazzi et al. (2019) aimed at carrying out a systematic analysis of the reliability and content of websites related to anorexia nervosa in the Italian language. Çelik et al. (2015) have shown a significant positive correlation between a problematic use of the Internet and eating disorders. A problematic use of the Internet is a predictor of eating disorders. Chang and Bazarova (2016) demonstrated that publications containing emotional words linked to stigma, the specific content of anorexia, and very correlational content generally trigger negative feedback from other members of the pro-anorexia community. Gale et al. (2015) have demonstrated that pro-eating disorder websites lead to preserving the behavior of the eating disorder. These websites seemed to offer a sense of support to teenagers with eating disorders. Hernández-Morante et al. (2015) have stated that pro-eating disorder websites influence eating behaviors, including obesity. Hilton (2018) has shed light on the role of pro-anorexia websites in eating disorders. Tan et al. (2016) have examined models of Internet and smartphone apps, for individuals showing eating disorders. Overall, any use of applications for smartphones was associated with a younger age and a higher psychopathology of eating disorders and psychosocial deficit. Yom-Tov et al. (2016) have explored the characteristics of people participating in different pro-anorexia web communities and the differences among them, and have shown that the women members of the main pro-ana website investigated seem to be depressed.

## 4. Discussion

According to the scientific literature, eating disorders are a serious psychiatric disease, with a mortality rate going from 5 to 6 percent, higher than all other mental disorders. In particular, anorexia is difficult to treat and, in the cyberspace, there is a phenomenon of pro-eating disorder websites, in particular pro-anorexia, that means there are many websites supporting anorexia—pro-ana and pro-mia. The aim of these websites is to promote an anorexic lifestyle. These websites can encourage unhealthy habits and in particular, can promote eating disorders [28,29]. Most analyzed research is focused on the social effects of pro-ana and pro-mia websites, and different authors have demonstrated that they most frequently are visited by the young female population [34,35,36]. However, some recent research suggests that pro-ana and pro-mia websites can encourage negative eating behaviors, such as promoting a pro-anorexic approach, through suggestions and tips to lose weight. The literature mainly examines the social influence of the media and has shown that these websites and blogs show ideal images of absolute thinness [37,38,39].

Several authors have identified many bloggers focused on pro-anorexic lifestyles and diets, giving advice and tips on how to lose weight (such as laxatives, purging in the shower, excessive exercise, calorie restriction, slimming pills, limitations in eating habits), extreme thinness, negative messages about food, and information about body image. Researchers have studied other aspects that can promote eating habits such as competition among members of these blogs to lose weight and be thin. Members of these communities would like to use their body image as inspiration models, for example, in the website’s gallery. This is in line with anti-recovery, because these blogs, communities, and/or websites reject recovery and medical treatment for eating disorders. Another form of resistance involved in these websites is disagreement with psychiatric and psychological treatment [30,31,32,33,34,35].

Most users of these websites are teenagers and have a complex psychological relationship with food—maybe they use it as a reward, a punishment, or they even feel guilty for eating some specific hypercaloric foods. According to the epidemiology of eating disorders, many accounts on pro-ana and pro-mia websites are managed by girls and in 58% of cases, websites contain images intended to encourage weight loss. Young women are particularly vulnerable to this kind of website, are attracted to them, and in fact, the most common people visiting pro-eating disorder websites are 13-, 15-, and 17-year-olds [10].

An important cognitive aspect found is connected to recursive thought. Sharing emotions is maladaptive when users, instead of reconsidering a negative event, continue to brood over it. Thinking continuously makes a negative event more enjoyable and prevents individuals from distraction, therefore intensifying stress, anxiety, shame, and other negative emotions associated with that event [40]. Brooding over a negative event and its meaning, without cognitively reconsidering that event, has proven to intensify negative emotions, making users even more depressed, anxious, and angry in comparison to the initial self-revelation. In these websites, users through pro-ana and pro-mia forums always brood over the same topic. Results have shown that mainly young women feel uncomfortable and often dissatisfied with their body image, and have consequently reduced their self-esteem and have difficulty in relational and social activities [41,42].

However, there is a limited number of empirical studies in the literature on the topic that have investigated the risks associated with the development of eating psychopathology in adolescents. The existing literature on pro-eating disorder websites has not focused on clinical and psychopathological behaviors that can lead also to a serious risk of suicide in young populations. It is very important to focus on the risk of using websites and forums encouraging maladaptive eating habits, especially in vulnerable teenagers with problems linked to food. Higher behavioral risks in the use of eating disorder websites show a higher tendency to isolation, negative emotions, and maladaptive eating habits, which can be predictors of eating disorders or can exacerbate subclinical pictures in vulnerable subjects.

The paper highlights the importance of a study focused on the psychological impact of pro-ana and pro-mia websites in female teenagers with vulnerabilities in eating habits. The psychological impact of images, texts, words, and of a maladaptive eating approach for an ultra-slim body is an extremely complex process, and is dangerous for teenagers’ body image. There are many psychological and identity development factors, as well as socio-cognitive skills and aspects of social desirability that are involved.

Our results suggest the importance of paying attention to psychoeducation among teenagers about eating disorders, food nutritional principles, and provide correct information on factors preparing, precipitating, or preserving the insurgence of eating disorders. This plays an important role in order to improve quality in their life and to reduce the risk of eating disorders.

## 5. Limitations

A limitation of the present study is the scarce literature on this topic. There are few empirical studies that explain the psychological motivation that attracts adolescents to eating websites.

## 6. Conclusions

As highlighted in the scientific literature, pro-ana and pro-mia websites promote a negative approach to food in a vulnerable population, such as in adolescents, and these conditions can encourage the insurgence of eating disorders.

Explaining this phenomenon connected to the use of forums and websites among female teenagers is fundamental because anorexia and bulimia are very serious and dangerous diseases, with higher incidence in the period of teenage development. However, it is very important to identify online contents and raise awareness on the level of danger of these websites and on maladaptive eating approaches among young people. Different studies have found out that a negative self-image can limit quality of life and pro-anorexia and pro-bulimia websites do not cause eating disorders but can encourage them. As far as the research topic is concerned, it is necessary to implement actions of promotion of guidelines and policies for healthy eating habits in the target population. Some aspects are relevant to improve the impact of research on prevention in the teenage population because at the moment, there are not many empirical studies in the literature explaining it.

The need to prematurely recognize maladaptive signs, words, beliefs, and approaches in adolescents for healthy eating guidelines and policies with specific programs of school education, even for teachers and parents. These healthy eating contexts can raise awareness on problematic eating behaviors and identify cases needing counselling and treatment or, in the most serious cases, hospitalization, therefore reducing the potential for a wide range of teenagers being trapped in the net. Further research to understand the correlation between personality profile and the impact of exposure to “Ana-mia” websites on the prevention of mental health issues in teenagers is recommended. The issue is relevant in young populations to prevent the risk of suicide and psychopathological and mental problems in adulthood.

## Figures and Tables

**Figure 1 ijerph-18-02186-f001:**
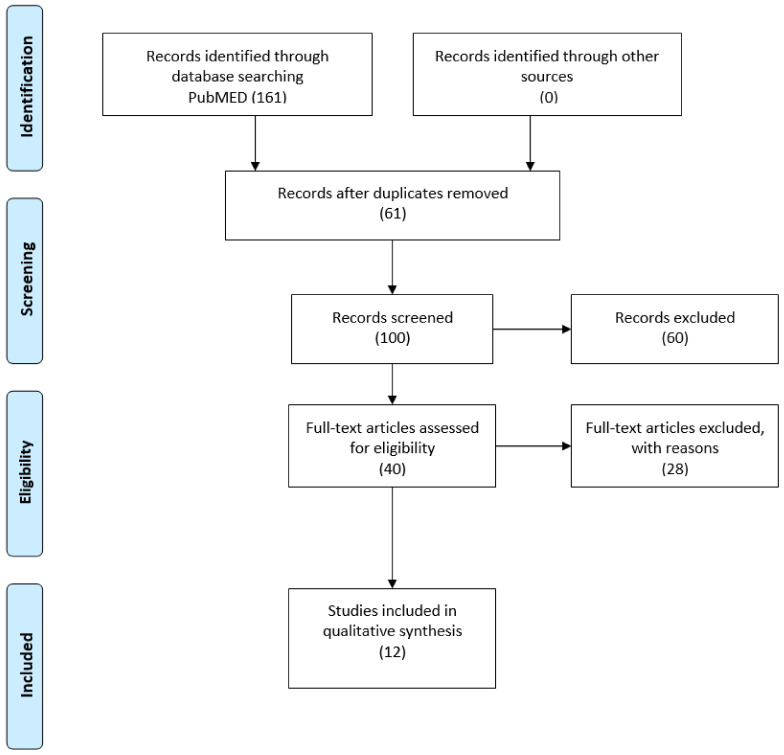
PRISMA (2009) flow diagram.

**Table 1 ijerph-18-02186-t001:** List of search terms entered into PubMed.

Number	Search Term
1	Pro (all fields)
2	Ana (all fields)
3	Blogging (all fields)
4	Mia (all fields)
5	1 AND 2
6	OR 3 AND 4
7	English (Language)
8	2015/01/01 to 2020/01/01 (publication date)

**Table 2 ijerph-18-02186-t002:** Characteristics of the studies included in the review.

Author	Aim	Sample	Type of Measure	Findings
Almenara et al., (2016)	This study explored the individual differences associated with adolescents’ exposure to ‘‘ana-mia’’ websites.	N = 18,709 girls, aged 11–16, 50%	-Exposure to ‘‘ana-mia’’ websites -Daily use of the Internet -Digital skills -Online disinhibition -Sensation seeking -Socioeconomic status of the household.	The results of this study showed that sensation seeking and online disinhibition were both associated with an increased risk of exposure to ‘‘ana-mia’’ websites in girls as well as in boys, although some gender differences were apparent. In girls, but not in boys, the older the child and higher the socioeconomic status, higher the chance of being exposed to ‘‘ana-mia’’ websites.
Bates (2015)	This study examined the metaphors the members of a pro-ana group invoked in their personal profiles on a popular social networking site, to talk about the self.	757 text profiles.	-The Metaphor Identification Procedure to 757 text profiles.	This study identified four key metaphorical constructions in pro-ana members’ self-descriptions: self as space, self as weight, perfecting the self, and the social self. These four main metaphors represented discourse strategies, both to create a collective pro-ana identity and to enact an individual identity as pro-ana.
Bert et al., (2016)	The aim of this study is to investigate the presence, popularity, and content of the ‘‘proana’’ accounts on Twitter.	341 accounts.	-Investigated the most used hashtags and the main contents of these profiles. -Twitter search. -For the statistical analysis of the retrieved data, used Stata MP11.	The authors found high number and popularity of Twitter proanorexia groups. These accounts contain dangerous information, especially considering the young age of the users.
Bragazzi et al., (2019)	The aim of the current investigation was to systematically perform a reliability and content analysis of Italian language anorexia nervosa-related websites.	402 unique website.	- Health on the Net Foundation Code of Conduct Standards (HonCode®) certification mark.	This study showed that the quality of Italian language anorexia nervosa-related websites was rather moderate-poor, being generally inconsistent with the principles of the HonCode^®^ certification mark.
Çelik et al., (2015)	The aim of this study was to investigate the relationship between problematic internet use and eating attitudes in a group of university students.	students.	-The Problematic Internet Use Scale -Eating Attitudes Test -Personal Data Form.	The research findings showed a significant positive correlation between problematic internet use and eating disorders, the problematic internet use is a predictor of eating disorders.
Chang & Bazarova (2016)	The aim of this study was to explore online negative enabling support dynamics in pro-anorexic websites through the language analysis of initiating disclosure and response sequences.	Analyzing 22,811 messages from 5590 conversations from the Pro-Ana Nation online discussion board forum.	- Linguistic Analysis.	The findings showed that initiating disclosures containing stigma-related emotion words, anorexia-specific content, and sociorelational content are typically met with negatively valenced responses from other members of the pro-anorexic community.
Gale et al., (2015)	The present study aimed to explore the underlying functions and processes related to the access and continued use of pro-ED websites within a clinical eating disorder population using a qualitative research design.	Seven adult women in treatment for an eating disorder who had disclosed current or historic use of pro-ED websites. Interviewees ranged in age from 20 to 40 years.	-Face-to-face semistructured interviews.	This study showed that Pro eating disorder websites maintaining eating disordered behaviour. This websites appeared to offer a sense of support for adolescents with eating disorder.
Hernández-Morante et al., (2015)	The objective of this study was to determine both general and information quality of eating disorder websites, including obesity websites.	50 websites.	- Three key terms (obesity, anorexia and bulimia) were entered into the Google® search engine. Websites were assessed using two tests (HonCode® certification and Bermudez-Tamayo et al. test) to analyze overall quality, and a third test (DISCERN test) to analyze specifically information quality.	This study determinated that the pro eating web sites influenced the eating disorders, including obesity.
Hilton (2018)	The aim of this study was to studied the Qualitative Exploration of the Role of Pro-anorexia Websites in User’s Disordered Eating.	151 members of pro-ana website.	- The analysis revealed five main themes: eating disorders are mental illnesses and websites do not cause mental illness, pro-ana websites and eating disorders.	This research revealed the role of pro anorexia websistes in eating disorders.
Tan et al., (2016)	The aim of this study was to assess a group of patients with eating disorders in Singapore who presented for treatment.	56 participants.	-Eating Disorder Examination Questionnaire 6.0 (EDE-Q 6.0) -Eating Attitudes Test26 (EAT-26) -Clinical Impairment Questionnaire 3.0 (CIA 3.0).	This study looked at the Internet and smartphone app usage patterns of participants who presented with an eating disorder in Singapore, and whether it corresponded to severity of illness. Overall, any smartphone application usage was associated with younger age and greater eating disorder psychopathology and psychosocial impairment.
Yom-Tov et al., (2016)	The aim of the current study is to explore the characteristics of people who participate in different pro-anorexia web communities and the differences between them.	Posts from the discussion board of the myproana.com website (A total of 57,911 post).	Identified users who used the terms in the 5 categories above. For those users, the most popular terms were categorized as follows: Myproana: users who queried for the myproana.com website. Tumblr: users who queried for the social network tumblr, Manorexia: users who queried for the term “manorexia”, meaning anorexia in males, Thinspiration, Yahoo Answers: visitors to the popular Yahoo Answers website.	Members of the main pro-ana website investigated appear to be depressed, with high rates of self-harm and suicide attempts, users are significantly more interested in treatment, have wishes of procreation and reported the highest goal weights among the investigated sites.
Yom-Tov et al., (2018)	The objective of this randomized controlled trial (RCT) was to examine if online advertisements (ads) can change online search behaviors of users who are looking for online pro-ana content.	10 different Bing Ads system.	-Using the Bing Ads system.	Exposure to the pro eating disorder websites, was associated an increased of eating disorders.

**Table 3 ijerph-18-02186-t003:** Prisma checklist.

Section/Topic	#	Checklist Item	Reported on Page #
Title			
Title	1	Psychological Impact of Pro-Anorexia and Pro-Eating Disorders Websites on Adolescent Females: A systematic review	1
Abstract			
Structured summary	2	The studies present in the literature showed that the websites defined as pro-ana and pro-mia supported eating disorders in female adolescents, because these websites promote an ideal of an ultra-thin body. The aim of this review is to explore the psychological risk of pro-ana and pro-mia websites in female adolescents. We carried out a systematic review of the literature on PubMed. We used the terms “Pro” AND “Ana” OR “Blogging” AND “Mia”. In the initial search, we identified 161 publications, and a total of 12 studies were selected according to inclusion criteria. The inclusion criteria were: English language, publication in peer reviewed journals, quantitative information on pro-ana and pro-mia websites among adolescents, in particular female adolescents, review articles, editorial comments, and case reports, and year of publication at least after 2015. We arbitrarily decided to start our research from 2015 to give a more recent view of the psychological impact of pro-ana and pro-mia websites on female adolescence findings. The PubMed database was searched from January 1 2015 to January 1 2020. According to the scientific literature, female adolescents were dissatisfied with their physical appearance, and the inappropriate use of the Internet increased the risk of eating disorders. It is very important to pay attention to this condition, to promote healthy eating habits, and to prevent the risk of eating disorders.	1, 3, 4, 5
Introduction			
Rationale	3	Many factors seem to influence the development and preservation of eating disorders among female teenagers. The scientific panorama shows how teenagers with eating disorders have a distorted body image, an inaccurate perception of their image, and therefore, are more dissatisfied with their physical aspect, in comparison to celebrities. There are a variety of pro-eating disorders communities (websites), and teenagers use social media to talk about their physical aspect, their activities, and to exchange advice about weight loss; this condition supports anorexia nervosa, as the problem of weight loss becomes relevant for their lives and the solution to their health problems. In fact, according to the scientific literature, all members of these communities have reported high levels of eating disorders.	2
Objectives	4	The aim of this work is to explore the psychological risk of pro-ana and pro-mia websites among female adolescents.	2
Methods			
Protocol and registration	5	We used these search terms: “Pro” AND “Ana” OR “Blogging” AND “Mia”. In the initial search, we identified 161 publications, but after, according to inclusion and exclusion criteria, we analyzed 12 studies.	
Eligibility criteria	6	We used these eligibility criteria: English language, publication in peer reviewed journals, studies about pro-ana and pro-mia websites and female adolescents, and year of publication at least after 2015. Articles were excluded by title, abstract, or full text for irrelevance to the topic. Exclusion criteria were: review articles, editorial comments, and case reports/series. We arbitrarily decided to start our research from 2015 to give a more recent view of “the psychological impact of pro-ana and pro-mia websites on female adolescents”.	3
Information sources	7	This systematic review was conducted according to Systematic Reviews guidelines. The PubMed database was searched from January 1 2015 to January 1 2020, using 4 key terms related to this topic (“Pro” AND “Ana” OR “Blogging” AND “Mia”).	3, 4
Search	8	Articles have been selected by title and abstract; the entire article was read if the title/abstract was related to the specific issue of the psychological impact of pro-ana and pro-mia on adolescents, and if the article potentially met the inclusion criteria. References of the selected articles were also examined in order to identify additional studies meeting the inclusion criteria.	4
Study selection	9	A comprehensive literature search was conducted in PubMed database, with the final search updated on January 2020. The initial search conducted used the keywords “Pro” AND “Ana” OR “Blogging” AND “Mia”. We used key terms related to the processes connected to the psychological impact of pro-ana and pro-mia websites or blogging.	4
Data collection process	10	The search of the PubMed database provided a total of 161 citations; no additional studies meeting the inclusion criteria were identified by checking the reference list of the selected papers. After adjusting for duplicates, 61 records were screened. Of these, 28 studies were excluded according to the inclusion and exclusion criteria. After the screening, a total of 12 studies assessing the processes connected to the psychological impact of pro-ana and pro-mia websites on adolescents met the inclusion criteria and were included in the systematic review.	4
Data items	11	Not specified.	
Risk of bias in individual studies	12	Across the included studies in this review, a potential database bias should be considered. Only articles written in English language were used, which might have compromised access to articles published in other languages.	5
Summary measures	13	“ana-mia’’ websites	1,2,3
Synthesis of results	14	Most of the research analyzed was focused on the negative impact of these websites on female adolescents, such as anorexia and bulimia.	5
Section/topic	#	Checklist item	Reported on page #
Risk of bias across studies	15	In all studies included in this review, a potential bias of the database should be considered. Only articles in English have been used, which could have compromised access to articles published in other languages.	5
Additional analyses	16	Not specified.	
Results			
Study selection	17	Record articles we identified on PubMed (n = 161); records identified through other sources (n = 0); records after duplicates removed (n = 61); records screened (n = 100); records excluded (n = 60); full-text articles assessed for eligibility (n = 40); full-text articles excluded; with reasons (n = 28); studies included in qualitative synthesis (n = 12).	Figure 1
Study characteristics	18	Almenara et al. (2016) studied the individual differences in adolescents on ‘‘ana-mia’’ websites, in a sample of N = 18,709 girls, aged 11–16, 50%. The authors used these types of measure: exposure to ‘‘ana-mia’’ websites; daily use of the Internet; digital skills; online disinhibition; sensation seeking; socioeconomic status of the household. Bates (2015) studied the metaphors used in pro-ana. Bert et al. (2016) studied 341 accounts on these websites. Bragazzi et al. (2019) studied the language in anorexia nervosa-related websites. Çelik et al. (2015) investigated the relationship between problematic internet use and eating attitudes in a group of university students. Chang and Bazarova (2016) explored online negative enabling support dynamics in pro-anorexic websites through language. Gale et al. (2015) studied pro-ED websites. Hernández-Morante et al. (2015): the objective of this study was to determine both general and information quality of eating disorder websites, including obesity websites; they studied 50 websites and three key terms (obesity, anorexia, and bulimia) were entered into the Google® search engine. Hilton (2018) studied the Qualitative Exploration of the Role of Pro-anorexia Websites in User’s Disordered Eating. Yom-Tov et al. (2016, 2018) explored the characteristics of adolescents who participate in different pro-anorexia web communities.	Table 2
Risk of bias within studies	19	In all studies in this review, a potential bias of the database should be considered. Only articles in English, which could have compromised access to articles published in other languages	
Results of individual studies	20	The recent scientific literature has identified a growing number of pro-ana and pro-mia blogs which play an important role in the etiology of anorexia and bulimia, above all in female teenagers. The feeling of discomfort and dissatisfaction with their physical aspect therefore reduces their self-esteem	5
Synthesis of results	21	Most analyzed research has focused on the negative influence of websites in female adolescents with eating disorders such as anorexia and bulimia. These websites have seemed to offer a sense of support to teenagers vulnerable to eating disorders. These studies have explored adolescent exposure to these websites, personal profiles related to access to social network, as well as pro-ana accounts on Twitter [18,21,26,28]. Other more social, aspects, linked to communication and language, have been explored in a recent study on language and information used on this website [29,30,31]. The relationship between a problematic use and abuse of the Internet and eating behaviors in adolescence has been investigated, as well as negative online support in the case of pro-anorexia websites [11,14,22]. Psychological aspects are generally explored as a potential risk of eating disorders to exacerbate or preserve symptoms of users’ eating disorders in a sample population [17,32]. Another study has explored the physical and mental state of people participating in pro-anorexia web communities [33]. In particular, Almenara and colleagues (2016) have demonstrated that looking for sensations and disinhibition online were both associated with a higher risk of exposure to ana–mia websites, in male and female teenagers, although some gender differences were evident. In girls, but not among boys, the older the teenager was and the higher her socioeconomic status was, the higher the chances were of being exposed to “ana-mia” websites. Bates (2015) identified four key metaphorical constructions in self-description by pro-ana members: self as space, self as weight, and improving the self and social self. These four main metaphors represented speech strategies, both in order to create a collective pro-ana identity and to enact an individual identity as pro-ana. Bert et al. (2016) highlighted a high number and popularity pro-anorexia groups on Twitter. These accounts contain dangerous information, especially considering the users’ young age. The investigation by Bragazzi et al. (2019) aimed at carrying out a systematic analysis of the reliability and content of websites related to anorexia nervosa in the Italian language. Çelik et al. (2015) have shown a significant positive correlation between a problematic use of the Internet and eating disorders. A problematic use of the Internet is a predictor of eating disorders. Chang and Bazarova (2016) have demonstrated that publications containing emotional words linked to stigma, the specific content of anorexia, and very correlational content generally trigger negative feedback from other members of the pro-anorexia community. Gale et al. (2015) demonstrated that pro-eating disorder websites lead to preserving the behavior of the eating disorder. These websites seemed to offer a sense of support to teenagers with eating disorders. Hernández-Morante et al. (2015) have stated that pro-eating disorder websites influence eating behavior, including obesity. Hilton (2018) has shed light on the role of pro-anorexia websites in eating disorders. Tan et al. (2016) have examined models of Internet and smartphone apps for individuals showing eating disorders in Singapore and checked if they corresponded to the severity of the disease. Overall, any use of applications for smartphones was associated with a younger age and a higher psychopathology of eating disorders and psychosocial deficit. Yom-Tov et al. (2016) have explored the characteristics of people participating in different pro-anorexia web communities and the differences among them, and have shown that the women members of the main pro-ana website investigated seem to be depressed.	8
Risk of bias across studies	22	In all studies included in this review, a potential bias of the database should be considered. Only articles in English have been used, which could have compromised access to articles published in other languages	5
Additional analysis	23	Not specified.	
Discussion			
Summary of evidence	24	Several authors have identified many bloggers focused on pro-anorexic lifestyles and diets, giving advice and tips on how to lose weight (such as laxatives, purging in the shower, excessive exercise, calorie restriction, slimming pills, limitations in eating habits), extreme thinness, negative messages about food, and information about body image. Researchers have studied other aspects that can promote eating habits such as competition among members of these blogs to lose weight and be thin. Members of these communities would like to use their body image as inspiration models.	8
Limitations	25	The limitations of this study were the poor empirical studies to describe these phenomena.	8
Conclusions	26	As highlighted in the scientific literature, pro-ana and pro-mia websites promote a negative approach to food in a vulnerable population, such as in teenagers, and these conditions can encourage insurgence of eating disorders. Explaining this phenomenon connected to the use of forums and websites among female teenagers is fundamental because anorexia and bulimia are very serious and dangerous diseases, with higher incidence in the period of teenage development. However, it is very important to identify online content and raise awareness on the level of danger on these websites and on maladaptive eating approaches among young people. Different studies have found out that a negative image of the self can limit quality of life and pro-anorexia and pro-bulimia websites do not cause eating disorders but can encourage them. As far as the research topic is concerned, it is necessary to implement actions of promotion of guidelines and policies for healthy eating habits in the target population. Some aspects are relevant to improve the impact of research on prevention in the teenage population because at the moment, there are not many empirical studies in the literature explaining it. The need to prematurely recognize maladaptive signs, words, beliefs, and approaches in a teenager can indicate healthy eating guidelines and policies with specific programs of school education, even for teachers and parents. These healthy eating contexts can raise awareness on problematic eating behaviors and identify cases needing counselling and treatment or, in the most serious cases, hospitalization, therefore reducing the potential for a wide range of teenagers being trapped in the net. Further research to understand the correlation between personality profile and the impact of exposure to “Ana-mia” websites on prevention of mental health in teenagers is recommended. The issue is relevant in young populations to prevent the risk of suicide and psychopathological and mental problems in adulthood.	7,8
Funding			
Funding	27	The author(s) received no financial support for the research, authorship, and/or publication of this article	8

*From:* (Mother et al., 2009) related to Prisma Checklist form [27].

## Data Availability

The study did not report any data.

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
