# Peer review of "Psychological Impact of Pro-Anorexia and Pro-Eating Disorder Websites on Adolescent Females: A Systematic Review"

_ijerph, 2021, doi:10.3390/ijerph18042186_

Round 1

Reviewer 1 Report

This is a valuable revision which aims to explorethe psychological impact of Pro Ana, and Pro mia Blog, among female adolescents.

Authors conducted a systematic review of 12 studies published from 2015 to 2020.

I have two main concerns whis are listed below:

  1. Results: Paragraphs should be analyzed by topics, such as psychological impact; members perception, linguistic analysis and metaphorical constructions; websites and social networks....
  2. Discussion should be more supported with previous studies and also analyzed by topics. Furthermore, I wonder why the Internet and social network still support these forums. There are policies which aim to avoid this kind of content.

Author Response

Please see the details in the attachment

Reviewer 2 Report

The paper submitted for review under the title "Psychological impact of Pro-Anorexia and Pro-Eating Websites in Adolescents female: A systematic review" seems to be very interesting, but it also has limitations.

Main remarks:
1. The authors present the purpose of the work: ,, In the light of data collected in scientific literature, the aim of this work is to explore the psychological impact of Pro Ana, and Pro mia Blog, among female adolescents ".
However, in the Conclusions section they also write about the quality of life, maybe it is also worth adding a quality of life assessment for work?
2. The second main consideration is the transparency of the work.
The authors provide tab 1 tab 2 in their manuscript as well as in Non-published Material. As the authors want to show the tables in manusctipt, they no longer show them in Non-published Material.
- please also consider from the text Table 3. Prisma Checklist.
Checklist should not appear in the body text.
3. Please return the limitation study of your research, because the authors write that they analyzed databases from 2015-2020, and only include 12 scientific articles in their analysis.

Additional notes:
- linguistic proofreading is recommended
- it is recommended to adapt the References section to the requirements of the International Journal of Environmental Research and Public Health
- please also remove the period from the title of the work (Psychological impact of Pro-Anorexia and Pro-Eating Websites in Adolescents female: A systematic review.)
- please standardize the size of letters in the names and surnames of the authors of the work
- lack of lines in the article makes the review difficult (please standardize the size of letters on page 6, regarding tables (References of the selected articles were also examined in order to identify additional studies meeting the inclusion criteria. Details are reported on Table 1 and table 2.) paragraph at the top of the page.

Author Response

(The authors gave the same response as above.)

Round 2

Reviewer 2 Report

Thank you for your improved version and the very thorough replies.

I have no further comments

This manuscript is a resubmission of an earlier submission. The following is a list of the peer review reports and author responses from that submission.